# Bruton’s Tyrosine Kinase Targeting in Multiple Myeloma

**DOI:** 10.3390/ijms22115707

**Published:** 2021-05-27

**Authors:** Max Von Suskil, Kazi Nasrin Sultana, Weam Othman Elbezanti, Omar S. Al-Odat, Robert Chitren, Amit K. Tiwari, Kishore B. Challagundla, Sandeep Kumar Srivastava, Subash C. Jonnalagadda, Tulin Budak-Alpdogan, Manoj K. Pandey

**Affiliations:** 1Department of Biomedical Sciences, Cooper Medical School of Rowan University, Camden, NJ 08103, USA; vonsuskim3@students.rowan.edu (M.V.S.); elbezanti@rowan.edu (W.O.E.); omaral98@students.rowan.edu (O.S.A.-O.); chitre72@students.rowan.edu (R.C.); 2Department of Chemistry and Biochemistry, College of Science and Mathematics, Rowan University, Glassboro, NJ 08028, USA; Jonnalagadda@rowan.edu; 3Department of Biosciences, Manipal University Jaipur, Jaipur 303007, India; kazinasrin1@gmail.com (K.N.S.); sandeepkumar.srivastava@jaipur.manipal.edu (S.K.S.); 4Department of Hematology, MD Anderson Cancer Center at Cooper, Cooper Health University, Camden, NJ 08103, USA; Budak-Alpdogan-Tulin@CooperHealth.edu; 5Department of Pharmacology and Experimental Therapeutics, The University of Toledo, Toledo, OH 43606, USA; Amit.Tiwari@utoledo.edu; 6Department of Cancer Biology, College of Medicine & Life Sciences, The University of Toledo, Toledo, OH 43614, USA; 7Department of Biochemistry and Molecular Biology, University of Nebraska Medical Center, Omaha, NE 68198, USA; kishore.challagundla@unmc.edu; 8The Fred and Pamela Buffett Cancer Center, University of Nebraska Medical Center, Omaha, NE 68198, USA; 9The Children’s Health Research Institute, University of Nebraska Medical Center, Omaha, NE 68198, USA

**Keywords:** multiple myeloma, microenvironment, Bruton’s Tyrosine Kinase (BTK) inhibitors, resistance, drug development

## Abstract

Multiple myeloma (MM), a clonal plasma cell disorder, disrupts the bones’ hematopoiesis and microenvironment homeostasis and ability to mediate an immune response against malignant clones. Despite prominent survival improvement with newer treatment modalities since the 2000s, MM is still considered a non-curable disease. Patients experience disease recurrence episodes with clonal evolution, and with each relapse disease comes back with a more aggressive phenotype. Bruton’s Tyrosine Kinase (BTK) has been a major target for B cell clonal disorders and its role in clonal plasma cell disorders is under active investigation. BTK is a cytosolic kinase which plays a major role in the immune system and its related malignancies. The BTK pathway has been shown to provide survival for malignant clone and multiple myeloma stem cells (MMSCs). BTK also regulates the malignant clones’ interaction with the bone marrow microenvironment. Hence, BTK inhibition is a promising therapeutic strategy for MM patients. In this review, the role of BTK and its signal transduction pathways are outlined in the context of MM.

## 1. Introduction

Multiple myeloma (MM) is clonal proliferation of malignant plasma cells in the bone marrow microenvironment (BMM), and it accounts for approximately 1% of all neoplastic diseases and 15% of all blood cancers, making it the second most common hematological malignancy after lymphoma [1,2]. There have been substantial changes in the treatment of MM during the last 20 years and those changes increased the five-year survival rates for patients from 12% to 54% (SEER data. Cancer Statistic Review 1975–2016. National Cancer Institute 2019). 

Plasma cells reside in the bone marrow where they normally comprise 0.25% of mononuclear cells, but in MM, monoclonal plasma cells accumulate and alter the BMM. This accumulation leads to disruptions in bone remodeling compartments, increased lytic activity, impaired cellular immunity and angiogenesis [3]. Historically, due to the rarity of plasma cells, their production and biology have been poorly understood. However, recent advances in the field have led to the identification of numerous potential therapeutic targets for MM [3]. These treatment modalities include combinations of proteasome inhibitors (PI), immunomodulatory drugs (IMIDs), alkylating agents, monoclonal antibodies (i.e., anti-CD38 and anti-SLAMF7), histone deacetylase inhibitors, B cell maturation antigen (BCMA) targeted drug-antibody conjugate, autologous stem cell transplantation (ASCT) and most recently BCMA targeting CAR-T cells. In spite of recent progress in the development of new and increasingly effective agents, MM remains incurable, mostly due to the evolution of drug-resistant clones [4]. Drug resistance of MM has been shown to be mediated through several mechanisms including loss of CD38 expression, BCMA expression, proteasome subunit β type 5 (PSMB5) point mutation or overexpression, and Cereblon (CRNB) down regulation, all of which help to produce MM stem-like cells (MMSCs). Therefore, successful alternative treatments of the stem-cell-like phenotypes is a critical focus in MM research moving forward [5,6].

Examining the molecular mechanisms of how the aggressive and drug resistant myeloma cells interact with the supporting stroma can allow for better patient tailored clinical management through targeting and disrupting the tumors’ protective niche [6]. 

## 2. Bruton’s Tyrosine Kinase, Structure and Regulation

Protein kinases (PTKs) act like a person flipping a molecular switch to control whether proteins are turned on or off. Kinases accomplish this by facilitating the transfer of a phosphate group from ATP to either a serine, threonine or tyrosine residue of their target protein [7,8]. BTK is one of the five Tec family kinases, a group of cytosolic kinases that phosphorylate tyrosine residues. The other four Tec family kinases include: Txk/Rlk, Itk/Emt/Tsk, Bmx and Tec, the enzyme which the family was named after [9]. The Tec family of kinases are one branch of the Src module kinases that mediate an array of cellular signaling processes. The other Src module kinases include ABL, CSK, FRK and BRK families [8]. 

Structurally, the SRC root module that the Tec family of kinases stem from consists of a single polypeptide chain comprising an SH3 and an SH2 domain. The SH2 domain is connected by an SH2-kinase linker to the kinase domain (AKA; SH1 domain), which is ultimately responsible for catalytic activity [8]. In BTK’s inactive state, the SH2 and SH3 domains impinge upon the kinase domain, thereby stabilizing inactivity and providing allosteric control over the catalytic site. The hallmark feature of the Tec family kinases that differentiates them from their relative kinases is the presence of the amino terminal Pleckstrin Homology (PH) domain that also confers allosteric control by further stabilizing the inactive conformation [9]. Following the amino PH domain is a proline-rich Tec homology (TH) domain, followed by the Src domains SH3, SH2 and the carbonyl SH1 domain [7] (Figure 1). 

Genetically, BTK has been shown in humans to be on the long q arm of the X chromosome at location Xq22.3-q22. The gene encompasses 37.5 kb, contains 19 exons and is expressed in all hematopoietic cells with the exception of T cells and possibly plasma cells [10]. Loss of the X chromosome has been reported in MM; however, these cases are leading to monosomy in females, therefore complete loss of BTK coding does not occur. There is little data supporting amplification of the BTK coding portions of the X chromosome in MM.

BTK is primarily cytosolic but can be recruited to the cell membrane due to interactions with its PH domain. The PH domain confers the ability to bind to phosphoinositides, thereby merging the tyrosine kinase and the phospholipid signaling systems [9,11]. BTK activation has been shown to be directly linked to its initial membrane recruitment and subsequent phosphorylation. Following its recruitment to the membrane, BTK activation occurs in two steps. Step one is the trans-phosphorylation of BTK on tyrosine residue at position 551 (Y551) in the kinase domain by either Lyn, SYK or other Src family kinases [12]. This first step is thought to be responsible for the enzymes’ catalytic ability and consequently results in the second step, autophosphorylation of tyrosine at position 223 (Y223) in the SH3 domain [13]. The Y223 position of the SH3 domain is highly conserved among signaling proteins and is thought to lie in the surface of the ligand-binding groove, which implies that this autophosphorylation may play a role in protein-protein interactions as well as stabilizing the active form, although the extent of these interactions is not fully understood [7,13]. 

BTK activation can occur through other Src kinases but is primarily regulated by its transient recruitment to the cell membrane through the canonical PH-TH module with phosphatidylinositol (3,4,5)-trisphosphate (PIP_3_) membrane lipids and subsequent phosphorylation. Little is known about the structural mechanism of this membrane interaction. However, recent simulations suggest that PIP_3_ acts as a regulator of BTK activity as it controls PH-TH Saraste dimerization within the cell membrane, which may be an important step in enzymatic activation [13,14,15]. BTK catalytic activity and tyrosine phosphorylation increases in response to cross linking or stimulation of the sIGM complex, the IL-5 or IL-6 receptors of B cells and the IgE receptor of mast cells, all of which are over expressed in MM patients. BTK activation then triggers activation of protein kinase B (AKT), phospholipase c (PLC) and calcium mobilization, leading to increased downstream effects including proliferation, differentiation and cell survival. Thus, chronic activation of BTK through the tyrosine kinase or phospholipid signaling systems can be a key driver in several neoplastic malignancies and a valuable therapeutic target in many more [16].

## 3. BTK as a Potential Target in Malignancies

The phosphorylation of proteins is by far the most common form of reversible post-translational modifications that modulate the activity of biochemical signaling pathways in humans [17]. The PTKs alter specific substrate activity and/or interactions with other proteins and are a fundamental method of controlling the cell cycle. PTK inhibition (PTKi) has proven to be an effective and accessible way to favorably alter the cellular signaling underlying hundreds of pathologies, including cancer [7,17]. This is largely due to the abnormal activation and dysregulation of these kinase mediated pathways which act as driving forces in many malignancies. In cancer cells, this abnormal activation can lead to increases in proliferation, stemness, survival, migration, motility and metabolism, all of which are associated with less favorable outcomes in patients [7].

The BTK pathway is universal in blood cells, as its signaling molecules are expressed in all hematopoietic lineages [18]. BTK in particular plays a crucial role in the development of peripheral B-cells as well as the survival of leukemic cells and their interaction with the BMM [7,19]. BTK is located in the cytoplasm of B-cells and is required for proper B-lymphocyte development, differentiation and signaling [11]. When the gene encoding BTK is mutated, a condition called X chromosome-linked agammaglobulinemia (XLA) results. This condition is characterized by minimal B cell populations and ultimately results from altered B cell receptor (BCR) signaling and high rates of apoptosis. Clinically, patients with XLA exhibit impaired immune function, reductions in all classes of serum immunoglobulins, markedly reduced B cell lymphocytes populations, severe neutropenia, and an inability to make antibodies in response to vaccines and repetitive bacterial infections that onset several months after infancy [20]. Due to this relation, the kinase itself is named after the Pediatrician Ogden Bruton, who reported the first case of XLA in 1952 (Figure 2) [21].

BTK plays an important role in various blood cell signaling pathways besides BCR signaling, including chemokine receptor 4 (CXCR4) signaling, toll-like receptor (TLR) signaling and Fc receptor signaling. Many of these signaling pathways result in common downstream effects due to the activation of BTK. For instance, BCR and CXCR4 activation of BTK have both been implicated to be important in controlling the adhesion of vascular cell adhesion proteins (VCAM) [7,22]. A comprehensive outline of how BTK interacts in each of these pathways and the implications it has in neoplasm is provided below.

B Cell Receptor Signaling: Due to BTK’s causal role in the B cell disorder XLA, the most thoroughly understood pathway which it mediates is B Cell Receptor (BCR) signaling [7]. In the germinal center, BCR, TLR and CD40 signaling are recognized as playing a fundamental role in the differentiation of B cells to plasma cells. After activation of B cells through BCR, TLR and/or CD40, upregulation of interferon regulatory factor 4 (IRF4) expression and downregulation of B-Cell Lymphoma 6 (BCL-6) expression occurs in addition to a loss of PR/Set Domain 1 (PRDM1) repression, resulting in down regulation of Paired Box 5 (PAX5) gene and upregulation of X-Box Binding Protein 1 (XBP1) and ultimately resulting in a loss of the B cell gene activation and phenotype. Furthermore, in the centrocyte region, stimulation of Interleukin 10, 21 or 6 (IL-10, IL-21 or IL-6) results in signal transducer and activator of transcription 3 (STAT3) activation, yielding PRDM1 overexpression and hence the gain of the plasma cell phenotype (Figure 3) [3].

BTK is essential in the B-cell receptor signaling of peripheral B cells, as it inhibits apoptotic signals and controls proliferation, differentiation, and adhesion. In the absence of BTK, peripheral B cells have a high rate of apoptosis, corresponding to BTK’s role in induction of anti-apoptotic protein Bcl-xL. Following BCR antigen engagement, either SYK or Lyn are required to initiate the formation of a signalosome at the cell membrane, allowing for efficient signal transduction. The signalosome also contains; Cbl-interacting protein of 85 kDa (CIN85), the SH2 containing leukocyte protein of 65 kDa (SLP65), and B cell linker (Blnk) [23]. SLP65 acts as a scaffold protein orientating several signaling proteins including SYK, BTK and phospholipase Cγ2 (PLCγ2). Through this complex, SYK phosphorylates BTK, thereby allowing BTK to activate PLCγ2 and initiate a cascade by activating four separate families of different non-receptor protein kinases, resulting in key cell proliferation and survival signaling events. These activated kinases include PLCγ2, mitogen-activated protein kinase (MAPK), protein kinase B (AKT) and components of the nuclear factor kappa-light-chain-enhancer of activated B cells (NF-κB) pathway (Figure 3). These kinases activate several notable downstream targets with direct impacts on increasing cell survival rate, drug resistance, stemness and ultimately relapse in the case of MM. 

After phosphorylation of PLCγ2 by BTK, PLCγ2 can hydrolyze phosphatidylinositol 4,5-bisphosphate (PIP2) into inositol triphosphate (IP3) and diacylglycerol (DAG). IP3 modulates intracellular calcium levels and thereby can activate nuclear factor of activated T-cells (NFAT) transcription through calcineurin and calmodulin. DAG mediates activation of PKCβ, which can induce activation of several members of the MAPK family, including extracellularly regulated kinases 1 and 2 (ERK1/2), c-jun N-terminal kinase (JNK), p38 and NF-κB (Figure 3). 

On the other hand, AKT activation by BTK leads to downstream activation of proliferation and pro-survival protein pathways such as mammalian target of rapamycin (mTOR), NFAT, Forkhead Box O proteins (FOXOs) and NF-κB. 

Besides the kinases it activates, the BCR has also been shown to play a role in controlling integrin α4β1 (VLA-4)-mediated adhesion of B cells to VCAM and fibronectin, alluding to the role BTK can play in the metastasis of B cell malignancies and their tumor-stroma interactions. BTK directly interacts with Wiskott-Aldrich syndrome protein (WASP), leading to its activation which induces RAC-dependent actin filament rearrangements. This has been shown to be dependent on BTK, PLCγ2 and PKCβ but not ERK1 or ERK2 [7,24].

Chemokine Receptor Signaling: Chemokines are secreted proteins that function as a leukocyte-specific chemoattractant. CXCR4 is one of the many chemokine receptors, and is a G coupled protein receptor expressed in many hemopoietic and non-hemopoietic cell types. CXCR4 has been studied intensively; it binds stromal cell-derived factor 12 (CXCL12), CD4 and CD74, it is overexpressed in 23 human cancers and plays important roles in immune response, hematopoiesis and pro-tumorigenicity. 

When CXCR4 binds its endogenous ligand, CXCL12, which is highly expressed in the bone marrow, the trimeric G-protein subunits of CXCR4 dissociate and activate several pathways involved in chemotaxis, intracellular calcium flux, cell adhesion, survival, proliferation and gene activation. Therefore, CXCR4 activation in neoplasm is linked to increased proliferation, metastasis, angiogenesis, drug resistance and ultimately relapse. Accordingly, blocking the receptor has been shown to have therapeutic effects in MM including the disruption of tumor-stroma interactions, reduced proliferation, sensitization to cytotoxic agents, impaired homing and inhibited metastasis [25,26]. Several studies have shown that in MM, CXCR4 is aberrantly activated and associated with the progression, spread and relapse of the cancer. CXCR4 activation in MM is shown to be induced by several factors including hypoxia, TNF-α, TNF-β and VEGF. 

BTK is an important signal transductor in CXCR4-CXCL12 signaling as BTK inhibition has been shown to deregulate CXCR4 expression and signaling [27]. After CXCL12 has bound to CXCR4, the Gα and Gβ subunits activate PI3K. PI3K activation then interacts with the PH domain of BTK, recruiting it to the cell membrane, allowing for BTK activation and leading to PLCγ2, AKT, MAPK and NF-κB pathway activation. In addition to PI3K activation of BTK, the Gα and Gβ subunits can both directly bind to BTK at the PH and TH domains and the Gα subunit can activate BTK directly. When BTK is inhibited, it has been shown to inhibit CXCL12 and CXCL13 induced phosphorylation of PLCγ2, ERK1, ERK2, JNK and AKT proteins, proving BTK’s role in this major pathway (see Figure 3) [7,28]. 

Toll-Like Receptor Signaling: Toll–like-receptors (TLRs) are membrane bound receptors that mediate the innate immune system and are activated by binding ligands produced by various pathogens [29]. Toll-like receptor (TLR) signaling has synergistic and crosstalk effects with CXCR4 and BCR signaling, all of which are mediated by BTK [30]. In TLR signaling, BTK is required for the downstream activation of NF-κB and interferon regulatory factor-dependent transcription of inflammatory cytokine and interferons due to BTK’s interaction with various proteins, including Myeloid differentiation primary response 88 (MYD88), Toll/Interleukin-1 receptor (TIR), Interleukin-1 receptor-associated kinase 1 (IRAK1) and Toll/Interleukin1 receptor domain-containing adaptor protein (TIRAP)/Myelin and lymphocyte protein (MAL) [7,28]. Following TLR binding to lipopolysaccharides, most TLRs bind MYD88, initiating a signaling cascade activating TIR, IRAK1 and TIRAP/MAL, leading to BTK activation. BTK’s interplay between TLR and BCR signaling may therefore interconnect the two pathways (Figure 3) [31]. 

Fc Receptor Signaling: Fc receptors are shown to be expressed broadly in hematopoietic lineages and can mediate phagocytosis, release of inflammatory mediators, antibody dependent cell-mediated cytotoxicity and humoral tolerance in different cell types. In plasma cells, FCγRIIB plays an important role in regulating survival and homeostasis [32]. Activation of FcγRI receptors results in the activation of BTK, STK and PI3K. On the other hand, FCγRIIB receptor activation inhibits BTK through the recruitment of phosphatases. Inhibiting Fc receptor pathways can lead to a reduction in the supportive role of macrophage cells and thus a reduction in vascularization.

BTK Inhibition in MM cells leads to decreases in several key effects of BCR, CXCR, TLR and Fc receptor activation. By blocking BTK and inhibiting all four of these pathways, MM cells causes decreased levels of phosphorylated ERK, AKT, and PLCγ2, yielding decreases in expression of many genes related to oncogenicity including mTORC1, NFκB, ELK1/2, NFAT, MYK and more (Figure 3). 

As previously touched upon, BTK activation through one pathway has important interactions and crosstalk with other pathway molecules. For instance, BTK is activated by the BCR pathway and then can indirectly affect components of the TLR and chemokine signaling pathway, thereby adding complexity to its study [28]. Due to BTK’s role in activating numerous malignancy related pathways such as NFκB, AKT, mTOR, NFAT and more, the compound appears as an extremely attractive target for a number of blood related malignancies. BTK is particularly suitable as a targeted therapy in MM due to its critical role in the development of B cells within the bone marrow and its effects on numerous components in the BMM (Figure 2).

## 4. Role of BTK in Bone Marrow Microenvironment

MM as a cancer ultimately resides within the BM, as this is the home of plasma cells. Over the past two decades it has become increasingly evident that alterations to the BMM and the signals arising from it have crucial effects in the progression of MM and several other B cell malignancies [3,33,34]. The treatment of MM is particularly difficult due to the supportive role which the BMM plays. The cellular compartment of the microenvironment consists of stromal cells, osteoblasts, osteoclasts, osteal-macrophages, endothelial cells, and immune cells. It has been shown that the BMM can aid in the differentiation, migration, proliferation, survival, and drug resistance of cancerous plasma cells [35]. The BMM produces its drug protective niche by releasing adhesion molecules and cytokines that allow hardy MM cells and stem cells to survive cytotoxic chemotherapies. Furthermore, when MM plasma cells and BM stromal cells (BMSCs) interact, it prompts the stroma to ramp up cytokine production further and activates several onco-related pathways including JAK/STAT, Ras/Raf/MEK/MAPK, PI3K/AKT and NF-κB (Figure 3) [36]. BTK has several important roles in these BMM pathways, which ultimately contribute to a substantial bulk of the BTK inhibitor’s therapeutic potential.

Accumulation of plasma cells in the BMM, as occurs in MM, interferes with osteoblasts activity, and disrupts the normal balance of osteoblast bone formation and osteoclast bone resorption. These interactions in the bone are referred to as bone remodeling compartments (BRCs). When MM disrupts BRCs, osteoblast count and overall function decreases, causing a net-loss of bone and the hallmark bone lesions found in patients. This effect is exacerbated by a type of positive feedback loop, since osteoclasts stimulate myeloma cell survival and proliferation through direct binding interactions and myeloma cells increase the number and activity of osteoclasts. Osteoclast activity and bone resorption both release growth factors, such as IGF-1, which activate the PI3K pathway phosphorylating AKT protein independent of the JAK/STAT pathway. IGF-1 can also induce MAPK phosphorylation. BTK has been shown to be an important part of PI3K signaling and its inhibition reduces PI3K signaling output, blunting some of these IGF-1 mediated effects. This is because BTK and PI3K interact in the same pathways and are both required to elicit full AKT activation [3,37]. Furthermore, BTK is selectively expressed in osteoclasts, leading it to have a direct role in osteoclast function and bone resorption. BTK has also been shown to mediate the migration of osteoclast precursors towards stromal cell-derived factor-1 (SDF-1) and their differentiation to osteoclasts [38]. 

IL-6 is overproduced mainly by the monocytes, myeloid cells, and stromal cells of BMM in patients with MM. This overproduction is triggered by the IL-1 produced by monocytes and myeloma cells, as only a small amount of IL-1 is required to promote the production of large amounts of IL-6. Patients with high serum levels of IL-6 are associated with poor prognosis [3]. In a study of MM cells in co-culture with BMSCs or osteoclasts, BTK inhibition has been shown to be most potent against IL-6 or stroma dependent MM cells, suggesting that cytotoxic effects against MM cells in these treatments was by proxy of altering the BMM. The cytotoxic effects mediated through the BMM were accompanied by the suppressing of multiple chemokines and cytokines, including IL-6, SDF-1, activin A, IL-8, BAFF, M-CSDF and MIP1B [39]. This may be attributed to BTK’s role in the NLRP3 inflammasome, which is known to be a key player in the activation of bioactive IL-1β. BTK was shown to be a direct regulator of said NRLP3 inflammasome via direct interactions. Without active BTK, the NRLP3 was unable to induce IL-1β secretion [40]. BTK has also been shown to have a strong effect on myeloid-derived suppressor cells (MDSCs) that inhabit the BMM, inhibiting T cell function and contributing to tumor progression. MDSCs are known to reduce the efficacy of immune based treatments and BTK inhibition has been shown to be able to inhibit MDSC generation, migration, and production of immunosuppressive chemicals such as nitric oxide (NO) and indolamine 2,3-dioxygenase [41].

BTK has also been implicated in playing a role in macrophages that reside and migrate to the BM. Emerging evidence shows that macrophages play an important role in primary tumors that metastasize to the skeleton. BTK inhibition in macrophages has been shown to suppress their production of CXCL12, CXCL13, CCL19 and VEGF, thereby negatively affecting the adhesion, invasion, and migration of lymphoid cells, including mechanisms not directly mediated through BTK [30]. The role in which macrophages play in the BMM in MM remains to be elucidated further but macrophages treated with BTK inhibitors have been shown to reduce their secretion of MM supportive factors and expression of NF-κB, STAT3 and AP-1 [41]. 

Besides its expression in the BMM, studies examining gene expression profiling and immunoblotting have demonstrated significant expression of BTK in more than 85% of MM patients. Studies in which BTK was blocked using Ibrutinib in myeloma cells showed inhibition of tumor survival and proliferation through the NFκB, STAT3, ERK1/2 and AKT signaling pathways [39]. The effect that BTK inhibitors have on chemokine receptors and fibronectin adhesion have been shown to have effects in myeloma cells and the microenvironment as well, resulting in tumors being released from various tissues into the blood away from their protective microenvironment niche, ultimately rendering them vulnerable [33]. Additionally, the overall homing of myeloma cells has been shown to be dependent on BTK [3,42]. This is significant because the adhesion of myeloma plasma cells to the BM stroma modulates refractoriness and is a trigger for epithelial to mesenchymal transition (EMT). CXCR4 has been shown to regulate this EMT change in MM as well as homing to the BM, indicating several methods in which BTK inhibition can sensitize MM cells by disrupting the interaction of plasma cells with the BM stroma [6].

BTK is highly expressed in the MM stem cell as compared to the bulk of MM cells. Overexpression of BTK in low expression cell lines increased stemness characteristics of cells such as clonogenicity, self-renewal and drug resistance. In particular, the AKT/Wnt/β catenin pathway dependent upregulation of key stemness genes NANOG, OCT4, SOX2 and MYC have been observed in myeloma cells overexpressing BTK (Figure 3). On the same note, inhibition of BTK in high expressing MM cell lines reduced these same characteristics and stemness gene expression [43,44].

BTK inhibition appears as a viable treatment option in the case of drug resistance, as BTK has been shown to be expressed and activated in certain drug resistant cells i.e., MM.1R, but not in parental non-resistant MM.1S cells, suggesting a shift in survival pathways [39]. Mechanistic studies have outlined that BTK activates the AKT pathway, leading to increases in drug resistance genes such as ABCB1 and BCL-2 and inhibition of GSK3β and in turn leading to activation of Wnt/β catenin pathways (see Figure 3), leading to drug resistance [43].

## 5. Development of BTK Inhibitors

Numerous BTK inhibitors have been developed over the past decade and a half, some of which have been FDA approved for the treatment of various B-cell malignancies (Figure 2) [45]. To date, no targeted kinase inhibitors have been approved for use in MM, despite effective results in other related conditions [46]. BTK inhibitors that have been approved by the FDA for other conditions include ibrutinib (PCI-32765) and acalabrutinib (ACP-196), both of which bind BTK irreversibly by forming a covalent bond to a cysteine residue at position 481 (C481) (Figure 4). The BTK-ibrutinib crystal structure had revealed residues playing important roles in the protein-drug interaction [47]. The SH moiety of C481, which is positioned close to the ATP pocket, forms covalent interactions with the head region of ibrutinib and allows the adenine ring atoms to form multiple hydrogen bonds with the backbone carbonyl of Glu475 and backbone amine of Met477 at the hinge region. These interactions thus orientate the diphenyl ether moiety to surround itself with conserved hydrophobic residues Leu542, Val416 and Met449 and charged residues like Lys430, Asp539 and Ser538 behind Thr474, which acts as a gatekeeper (Figure 4). 

Ibrutinib was the first developed BTK inhibitor and had been shown to have a large amount of off target activity, associated side effects and resistance due to mutation at the nucleotide binding pocket, thereby affecting the binding of inhibitors of this class [7,48,49,50]. Due to these issues, researchers have since taken unique approaches to develop a diversity of inhibitors that are even more selective and that reversibly bind to the SH3 pocket of the enzyme [51,52,53]. 

Currently, there are close to 20 BTK inhibitors undergoing different phases of clinical trials for numerous B-cell related malignancies. These different drugs include compounds which, depending on their structure, can selectively bind BTK irreversibly at either the C481 or Y223 position or reversibly bind at the C481 position. Said C481 position binding may improve binding selectivity and efficacy of these second-generation inhibitors (Figure 5). Some notable inhibitors currently in or recruiting for clinical trials for B-cell malignancies in addition to ibrutinib (PCI-32765) and acalabrutinib (ACP-196) which are both undergoing phase 1–3 trials include zanubrutinib (BGB-3111) a highly selective inhibitor which binds irreversibly at C481 and is undergoing phase 1–3 trials, vecabrutinib (SNS-062) which binds both highly selectively and reversibly at C481 as well as tirabutinib (ONO/GS-4059) which binds irreversibly at the Y223 position. Some BTK inhibitors are undergoing trials for autoimmune diseases as well, which may hold promise [7,49,54,55,56].

Side effects of BTK inhibitors are still being discovered but some major ones appear to be cardiac related. This is not surprising considering the role that BTK plays in B-cell development, but information about it remains insufficient. Atrial fibrillation seems to be the primary cause for concern regarding BTK inhibitor side effects. Additionally, interactions between BTK inhibitors and both anticoagulants and antiarrhythmic agents have appeared [57]. Increased risk of infection, neutropenia, hypertension, hair and nail changes, rash, diarrhea, and fatigue have all been reported to occur at least semi-frequently and should be considered with monotherapy of BTK inhibitors. Cataracts have been observed in animal studies and issues with bleeding and bruising have been described in studies combining BTK Inhibitors with anticoagulant/anti-platelets [58,59]. These side effects should be considered in study designs to reduce potential harm to at risk patients with co-morbidity. 

There are still many exciting BTK inhibitors in the drug development pipeline. These drugs in preclinical development include most notably compounds such as RN-486, CGI-1746, CNX-774 and LFM-A13 (Figure 6). Compound RN-486 is a small, potent, reversible BTK inhibiting molecule with demonstrated highly specific sub-nanomolar activity. It is mainly being examined in preclinical trials for autoimmune conditions such as rheumatism and lupus [60,61]. CGI-1746 is a unique, extremely selective, ATP-competitive, small molecule inhibitor of BTK. This molecule remarkably binds in a way to fill the SH3 binding pocket within the inactivated BTK molecule, thereby stabilizing it. It inhibits both auto- and trans-phosphorylation of BTK but has yet to progress past preclinical trials in mice [61]. CNX-774 is an irreversible, small molecule BTK inhibitor that is orally bio-available. It is extremely selective for BTK, as it forms a ligand directed covalent bond within the ATP binding site at the Cys-481 position. CNX-774 is undergoing advanced preclinical trials [61]. LFM-A13 is another unique BTK inhibitor considered to be a first in class molecule due to its dual BTK/Polo-like kinase inhibition. It is noted as having extremely promising effects in multiple cancer cells, including lymphoma and leukemia. In mice, LFM-A13 has had similar effects on myeloma cell growth, homing to bones and resorption, and the same is true of other BTK inhibitors such as ibrutinib [61].

Despite its efficacy, many patients have had to discontinue ibrutinib due to resistance and subsequent disease progression. Some ibrutinib resistant patients have been shown to have a C481S point mutation in BTK, as ibrutinib irreversibly binds to the C481 residue (Figure 4), leading to the drug only reversibly inhibiting the mutated protein [62]. The predominant strategy of covalently blocking C481 in inhibitor design and the subsequent emergence of drug resistance due to C481S mutants’ clonal selection highlight the importance of adopting diverse strategies of binding as well as reversible and irreversible binding mechanisms to further develop BTK inhibitors. Furthermore, the ability to bind and inhibit BTK at different sites may also confer an advantage for avoiding mutations. 

Gain-of-function mutations have been identified as well in the treatment of CLL using ibrutinib, causing BCR signaling to continue functioning normally despite BTK inhibition, thus highlighting the important role that BCR signaling inhibition has on the effect of BTK inhibition [63]. However, since the C481S mutation had been identified, in patients’ refractory to ibrutinib, other BTK kinase domain mutations have been identified including C481F/Y/R, T474I/S and L528W. Additionally, in CLL patients, mutations in BTK’s SH2 domain occurs at a frequency of 75% as well. Unlike mutations in kinase domain that impair ibrutinib binding, SH2 domain T316A is far away from the ibrutinib binding site, yet still confers resistance to a similar extent as C481S [64]. The significance and role of these other mutations in the kinase domain and SH2 domain are still under investigation. 

## 6. How Good a Target Is BTK for Myeloma Therapy?

Targeting the BTK pathway stands as another novel means of treating and further extending the lifespan of patients battling MM. BTK inhibition has a particularly important treatment potential because of its multifaceted effect. Inhibiting this one pathway can disrupt several aspects of the vicious cycle in which MM cancer cells and the BMM interact. BTK inhibition has five primary effects in myeloma therapy. First and foremost, BTK inhibition directly inhibits MM tumor growth. Second, it inhibits osteoclastic bone resorption, thereby increasing tumor cytotoxic potential. Next, it inhibits the release of osteoclast-derived tumor growth factors, further reducing tumor growth ability. Fourth, BTK inhibition prevents the release of BM stromal cell derived growth factors, thereby mitigating MM cell adhesion to the BM stromal cells [65]. Lastly, BTK inhibition impairs MM stemness, as BTK knockdown reduces several key MM stem cell characteristics such as side population cells, clonogenicity, induced pluripotent/embryonic stem cell genes and drug resistance [44]. Therefore, BTK is unique, as it has been said to attack “the seed and soil” [39]. Although it has potent effects on MM tumor and stem cells directly (the seed), what makes this agent so powerful is the effect it has on the supportive BMM niche (the soil).

There have been several clinical trials completed using BTK inhibition in MM as either a monotherapy or part of a combination therapy. Below is a brief outline of the different clinical trials completed and key points that can be taken from them.

BTK inhibitors used in combination drug therapies have seen very promising results in patients with R/R MM. Ibrutinib has been shown in patients with R/R MM to be very effective in combination with low dose dexamethasone. In the clinical trial NCT01478581, patients with R/R MM who have previously received at least two lines of therapy with one of which being an immunomodulator, were given escalating dosages of ibrutinib daily with or without 40 mg of dexamethasone weekly in a Simon Stage 2 study design. Despite being a heavily treated population with patients having received a median of 4 prior treatments, ibrutinib with dexamethasone still produced encouraging responses [46]. The primary outcome of clinical benefit response (CBR) showed that 420 mg/day of ibrutinib alone had a 7.7% CBR with a duration of 27.6 months. Additionally, a higher dosage of 840 mg/day of ibrutinib with 40 mg of dexamethasone per week had a 27.9% CBR with an average duration of 11.5 months. All patients completed the trial, highlighting the notable safety profile of ibrutinib and likely other BTK inhibitors. 

In phase 2 clinical trial NCT02943473, ibrutinib was also examined in patients with high-risk smoldering (HRS) MM. The patients received 560 mg of ibrutinib daily and primary outcome showed that 71.4% of patients remained free of MM for at least 1 year. However, the trial was terminated and none of the secondary outcomes had data reported. A total of 50% of patients with HRS MM are expected develop MM 2 years after diagnosis. A better controlled study with larger sample sizes and higher dosages of ibrutinib may yield more encouraging results.

The BTK inhibitor acalabrutinib was recently examined with and without dexamethasone in a phase 1b study of patients with MM. In one arm, patients received 100 mg of acalabrutinib twice per day and showed a minor response (MR) frequency of 7.7%, a stable disease (SD) frequency of 38.5%, a progressive disease (PD) frequency of 30.8% and a not evaluable (NE) frequency of 23.1%. In the other arm, patients received 100 mg twice per day with 40 mg dexamethasone once weekly and showed a MR frequency of 14.3%, SD frequency of 42.9%, a PD frequency of 7.1% and a NE frequency of 35.7%. These results led to the study being terminated due to lack of efficacy. However, valuable insights regarding the safety profile and pharmacokinetics of this BTK inhibitor were obtained. 

BTK inhibition and bortezomib as a combination therapy has also shown promising results, as both isolate the NF-κB pathway in different yet synergistic ways. Additionally, BTK therapy has been shown to sensitize bortezomib resistant MM cells to bortezomib [66]. Clinical trial NCT02902965 examined this combination using ibrutinib with the addition of dexamethasone in a phase 2 study treating R/R MM patients who had received 1–3 prior lines of therapy and had demonstrated progressive disease since last treatment. However, the study was suspended following increased incidence of infections. Incidence of adverse events were however largely consistent with previous reports across hematologic malignancies and the study was restarted. Nevertheless, 57% of patients had clinical responses with a median response duration of 9.5 months. The study was ultimately terminated due to an inability to achieve the primary endpoint [67].

Clinical trial NCT01962792 examined BTK inhibition in combination with carfilzomib with and without dexamethasone in R/R MM patients via a phase 1/2b study. All patients completed the dose escalation and DLT observation portion of this study. Patients received a median of 3 prior therapies, including 70% that received autologous stem cell transplants. The phase 2b portion used either 560 mg or 840 mg of daily Ibrutinib with IV carfilzomib and dexamethasone. A total of 30% of patients discontinued treatment due to toxicity. The study indicated promising results, despite most patients having advanced MM and a large subset being refractory to bortezomib and an IMiD. The study yielded an ORR of 67%, CBR of 76% and median PFS of 7.2 months. Remarkably, patients that were refractory to bortezomib showed particularly robust responses, with an ORR of 71% and median DOR of 9.1 months [68].

The clinical trial NCT02548962 used ibrutinib with pomalidomide and dexamethasone in R/R MM patients. The trial was terminated by the sponsor after completing phase 1. Ibrutinib was given in either 420 mg or 840 mg in a combination with 4 mg of pomalidomide and 40 mg of dexamethasone. The group that received 420 mg of ibrutinib with pomalidomide and dexamethasone showed a 50% CBR frequency and average 6.5 -month DOR. The group that received 840 mg of ibrutinib in the combination had a 66.7% CBR frequency and 7.3-month DOR.

Despite all the advances in treatment modalities, MM remains incurable. For this reason, there is a need to continue researching and expanding options for long term disease treatment and relapse control in MM patients. No matter how effective the treatment, MM relapses begin occurring more frequently after each remission. Therefore, targeting the myeloma stem cell appears to be a must. BTK has been shown to be one of the most promising kinase inhibitor options currently in development for MM. This is due to BTK’s robust expression in MM cells and its effect on both the tumor, stem cells and BMM [43,69]. BTK inhibition does not appear to be as cytotoxic to tumor cells as other treatments such as proteasome inhibitors or immunomodulators; however, the effect of targeting the MM stem cells and BM microenvironment may prove it to be an even more valuable and effective treatment in the long term and in relapsed/refractory (R/R) patients. This may prove to be especially important because the current treatments available still fail to prevent patients from relapse [70]. 

## 7. Future Directions

BTK inhibition has shown remarkable activity across B-cell histologies and has promise to be used as a single drug therapy for MM patients in certain circumstances due to its efficacy and lack of toxicity [39,65,71]. BTK inhibitors appear to be most valuable in isolation therapy for cases of drug resistant MM, as the BTK pathway appears to be selectively upregulated in these phenotypes [72], and in preventing relapse or disease progression. 

Combination therapies including BTK inhibitors have strong evidence of efficacy via the several clinical trials completed. Combination with immune therapies such as monoclonal antibodies have become an area of interest for researchers as well. Given BTK inhibitors’ method of interaction with the bone marrow and stem cells, combination therapies appear to be an extremely promising direction for future research [45,66]. 

## 8. Conclusions

BTK inhibitors appear to be an exciting and viable new approach within the expanding arsenal of drugs used to treat MM. A diverse set of irreversible and reversible BTK inhibitors varying in shape, size and chemical scaffolds have been developed over the years. Structural details of these inhibitor complexes reveal conformational plasticity of the BTK nucleotide binding pocket, which could be exploited further to explore a rich set of pharmacophores for structure-based design of inhibitors relevant to MM phenotypes. In the kinase pocket, structural flexibility of the helix C, Gly-rich loop, DFG sequence and activation loop offer diverse approaches to design novel drug leads based on the physiological relevance of these states (Figure 5). Our analysis of diverse ensembles of BTK-inhibitor structures show conserved active site waters whose spatial disposition can also be utilized to design/add novel scaffolds to mimic interactions of displaced water oxygens (Figure 6). 

Inhibiting the BTK mediated pathways offers a variety of benefits against both MM cancer cells themselves and the tumor microenvironment niche, which support the cancers by promoting growth and resistance to treatment. Additionally, the effect that BTK inhibition has on myeloma cell homing and adhesion to the bone marrow can inhibit important functions in MM survival, reducing cytokine secretion and sensitizing drug resistant cells to chemotherapy [73]. With all factors considered, BTK inhibition in MM has an irreplaceable three-pronged effect, attacking the cancerous plasma cell itself, its stem cells and the BMM, presenting it as one of the most exciting and robust currently developing MM targeted therapies.

## Figures and Tables

**Figure 1 ijms-22-05707-f001:**
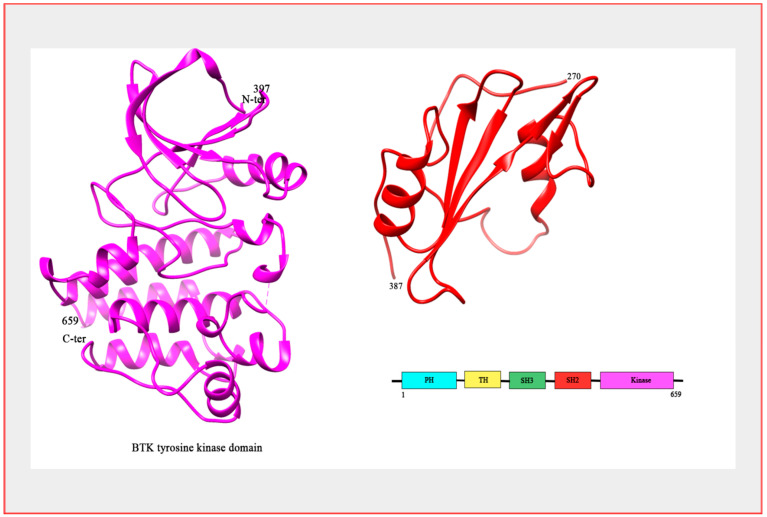
Structure of BTK. The crystal structure of BTK kinase domain (colored magenta; PDB: 3GEN) and SH2 domain (colored red; PDB: 2GE9). Bar Diagram representing the domain organization of the BTK protein.

**Figure 2 ijms-22-05707-f002:**
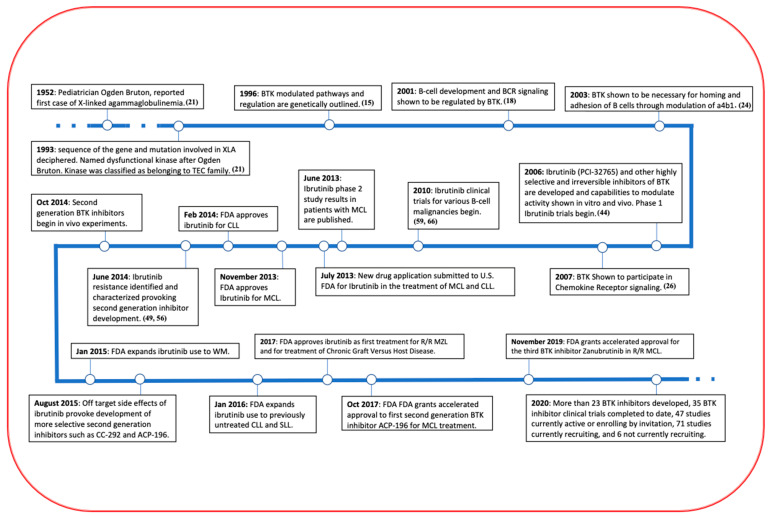
A Timeline of BTK and its inhibitor. BTK has made remarkable progress in the past 10 years in its treatment in a variety of B cell related malignancies and even more clinical trials are underway, including some in MM.

**Figure 3 ijms-22-05707-f003:**
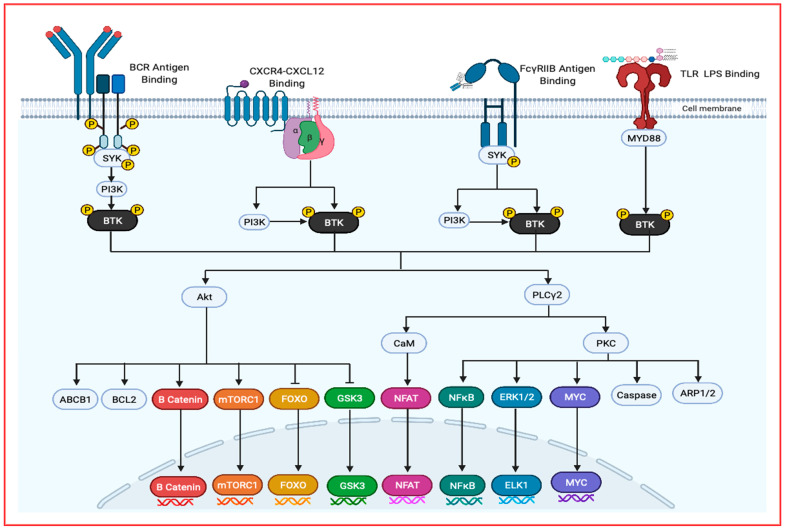
An overview of BTK activators and downstream signaling targets. BTK can be activated through several pathways include BCR signaling, chemokine signaling, Fc receptor signaling and TLR signaling. Following BTK integration to the cell membrane through its PH domain interaction with PIP3, the phosphorylated protein becomes cytosolic. These signaling pathways lead to the activation of AKT, PLCy2 and a protein complex including BTK, IRAK(1,2,4) and more leading to the downstream transcription factors that promote cancer growth, survival, stemness and migration.

**Figure 4 ijms-22-05707-f004:**
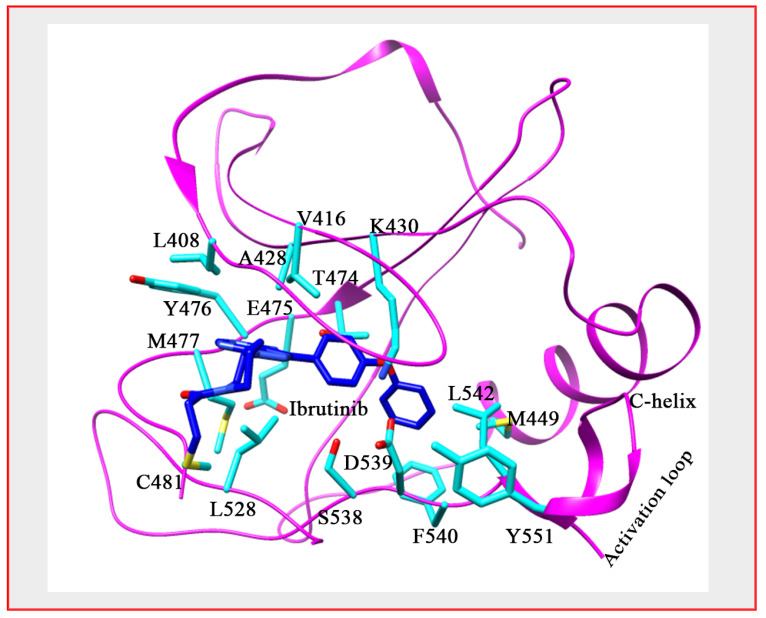
BTK’s interaction with ibrutinib. The crystal structure of BTK bound with inhibitor ibrutinib showing the interaction with protein (PDB: 5P9J). Protein colored in magenta, side chain in cyan and Ibrutinib in blue, ball and stick.

**Figure 5 ijms-22-05707-f005:**
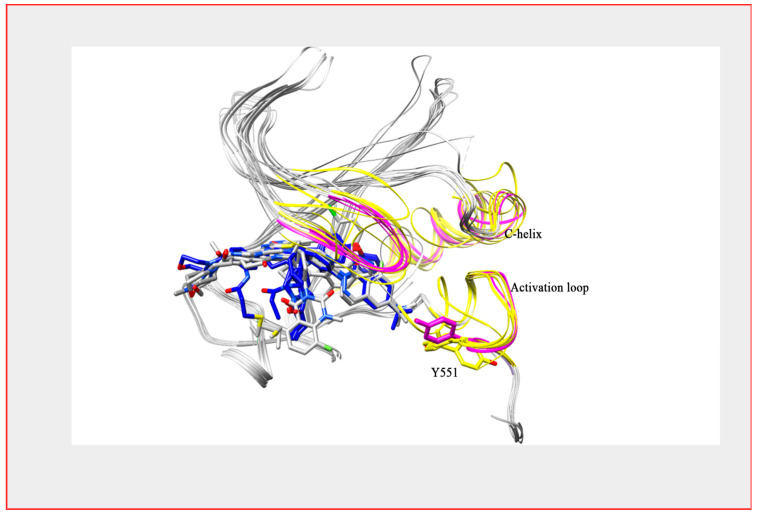
BTK structure with reversible and irreversible inhibitors. Superimposed structure of BTK kinase domain bound with both a reversible (colored grey) and irreversible inhibitor (colored blue). Secondary structure differences are shown in yellow (reversible) and magenta (irreversible) colors. The structure shows a structure difference at the C-helix and activation loop region.

**Figure 6 ijms-22-05707-f006:**
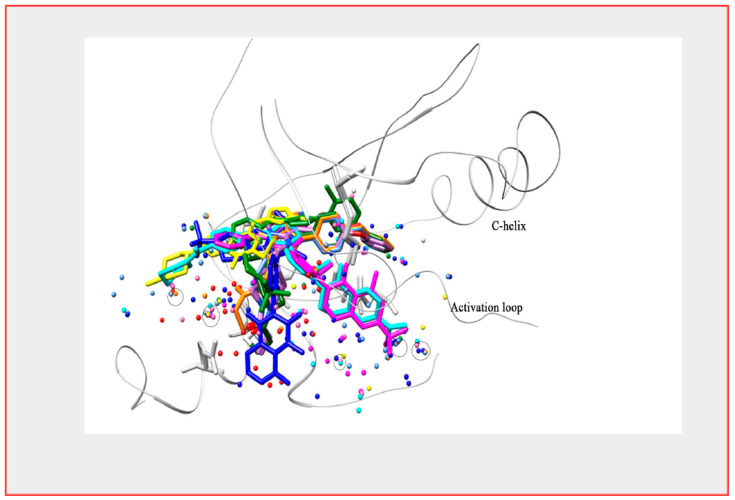
BTK and its interaction with reversible and irreversible inhibitors. Ten different reversible inhibitors (PDB: 3GEN, 3K54, 3PIX, 3PIZ, 4RFY, 4Z3V, 5KUP, 5T18) and irreversible inhibitors (PDB: 5P9J, 6J6M, 6OMU, 6N9P) are shown bound to a superimposed crystal structure of the BTK kinase domain. The figure focuses on the conserved (circled) and invariant water molecules at the active site.

## Data Availability

Not applicable.

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
