# Peer review of "Bruton’s Tyrosine Kinase Targeting in Multiple Myeloma"

_ijms, 2021, doi:10.3390/ijms22115707_

Round 1

Reviewer 1 Report

Very well written and organized manuscript. I only have few minor comments:

The chromosome X is commonly altered in myeloma, some part are found deleted and others gained. Is BTK gene copy number affected consequently?

Line 450-453, what is the % of high risk SMM patients that remained free of myeloma for at least a year in the control group (without therapy).Remind us please.

Line 492, typo 840mg instead of 8400mg

Line 502, you mean "proteasome inhibitors" not protease.

Author Response

Response to Reviewer 1 Comments

Comment #1:  The chromosome X is commonly altered in myeloma, some part are found deleted and others gained. Is BTK gene copy number affected consequently?

Response 1: This is really a good question. Mutations alter the gene function. However there is no studies whether common translocations of myeloma results in mutation or gain of function of BTK. Thus, we did not add in the text.

Comment #2: Line 450-453, what is the % of high risk SMM patients that remained free of myeloma for at least a year in the control group (without therapy).Remind us please.

Response 2: The 50% of those with High-Risk Smoldering MM that were not receiving treatment will develop MM in two years after diagnosis. However no data was provided for 1 without treatment.

Comment #3: Line 492, typo 840mg instead of 8400mg

Response 3: Thanks for finding the error, we have fixed it now. Thank you!

Comment #4: Line 502, you mean "proteasome inhibitors" not protease.

Response 4:  We have fixed it now. Thank you!

Reviewer 2 Report

ijms-1219394

Bruton’s Tyrosine Kinase Targeting in Multiple Myeloma.

The article " Bruton’s Tyrosine Kinase Targeting in Multiple Myeloma. " (ijms-1219394) by Von Suskil M et al. demonstrated that BTK played an important role for anti-B cell malignancies, including myeloma, treatment via mode of direct action for malignant cells and indirect action for bone marrow microenvironment. These data supposed to be very informative and suggestive for learning the role of BTK for MM treatment. However, there are several minor issues to be addressed as below.

Minor issues

  1. This article was very informative, but the volume of the article was too much. I considered that the session “2. Bruton’s Tyrosine Kinase, Structure and Regulation” and “Table 1. Inhibitors of Bruton’s Tyrosine Kinase” should be shortened if you can.

  1. I considered that “Fig3. An overview of BTK activations and downstream signaling targets” was not enough to explanation for BTK signaling. The author described the BTK was one of important therapeutic target for malignancies including myeloma in chapter 3 and 4 in detail, and These contexts were very informative. First, the author should show the figure about BTK signaling categorized into “B cell receptor signaling”, “chemokine receptor signaling”, “Toll-like receptor signaling”, “Fc receptor signaling” similarly with main documents. Next, the author should show the figure about role of BTK in bone marrow microenvironment including bone formation, immune system, and cytokine production.

  1. The author could add the mechanisms about resistance for BTK inhibitors at the end of chapter 5 “Development of BTK inhibitors”.

  1. Although “Table 1. Inhibitors of Bruton’s Tyrosine Kinase” was too much, little data was shown about clinical trials for myeloma. The author could decrease the volume about the other malignancies and focus about BKT inhibitors treatment for myeloma including summary of clinical outcomes which was shown in main document.

Author Response

Response to Reviewer 2 Comments

Comment #1: This article was very informative, but the volume of the article was too much. I considered that the session “2. Bruton’s Tyrosine Kinase, Structure and Regulation” and “Table 1. Inhibitors of Bruton’s Tyrosine Kinase” should be shortened if you can.

Response 1: Authors are thankful for encouraging words reviewer’s suggestions. We have now removed the table 1 and shorten the “Section 2. Bruton’s Tyrosine Kinase, Structure and Regulation”.

Comment #2: I considered that “Fig3. An overview of BTK activations and downstream signaling targets” was not enough to explanation for BTK signaling. The author described the BTK was one of important therapeutic target for malignancies including myeloma in chapter 3 and 4 in detail, and these contexts were very informative. First, the author should show the figure about BTK signaling categorized into “B cell receptor signaling”, “chemokine receptor signaling”, “Toll-like receptor signaling”, “Fc receptor signaling” similarly with main documents. Next, the author should show the figure about role of BTK in bone marrow microenvironment including bone formation, immune system, and cytokine production.

Response 2: We have modified the figure and added more information about the IL-1, CXCR4. The focus of the review is to provide detailed information about the BTK inhibitors and new developments. BTK plays critical role in bone marrow microenvironment, including bone formation, immune system, and cytokine production, we would like to cover this toic separetly in next review, which we are working already.  

Comment #3: The author could add the mechanisms about resistance for BTK inhibitors at the end of chapter 5 “Development of BTK inhibitors”.

Response 3: We have provided now resistance information at the end of the chapter as suggested.

Comment #4: Although “Table 1. Inhibitors of Bruton’s Tyrosine Kinase” was too much, little data was shown about clinical trials for myeloma. The author could decrease the volume about the other malignancies and focus about BKT inhibitors treatment for myeloma including summary of clinical outcomes which was shown in main document.

Response 4: We agree with reviewer’s suggestion. The table had little to do with BTK inhibitors in MM; therefore, it is removed from the modified version.